# Development of a SPE-LC-MS Method for the Quantitation of Palbociclib and Abemaciclib in Human Plasma

**DOI:** 10.3390/molecules27238604

**Published:** 2022-12-06

**Authors:** Daniela Maria Calucică, Costel-Valentin Manda, Amelia Maria Găman, Ștefan Răileanu, Liliana Stanca, Monica Denisa Elena Popescu, Olivia Garofița Mateescu, Andrei Biță, Octavian Croitoru, Simona-Daniela Neamțu

**Affiliations:** 1Faculty of Pharmacy, University of Medicine and Pharmacy of Craiova, 200638 Craiova, Romania; 2Faculty of Medicine, University of Medicine and Pharmacy of Craiova, 200638 Craiova, Romania; 3Clinic Oncology Municipal Hospital “Filantropia”, Filantropiei Street No. 1, 200638 Craiova, Romania

**Keywords:** palbociclib, abemaciclib, human plasma, SPE, LC-MS

## Abstract

Palbociclib and abemaciclib are two cyclin-dependent kinases 4 and 6 used for breast cancer treatment. Levels of these medicines present a significant interindividual variability, so monitoring those concentrations might be necessary in therapy. Most of the methods presented so far in the literature use simple protein precipitation of plasma proteins as sample preparation method followed by direct injection of the supernatant into the LC instrument, preceded or not by a simple filtration step. Within that approach, the probability of injecting proteins in the chromatographic system is increased. With the purpose of obtaining a cleaner extract of the drugs, we developed and validated a simple and accurate LC-MS method for determining palbociclib and abemaciclib in human plasma. Solid phase extraction (SPE) using Oasis PRiME HLB^®^ cartridges was used for plasma sample preparation. The method provided clean extracts with a recovery extraction higher than 85% for both compounds. Separation was achieved by high-performance liquid chromatography (HPLC), using a C18 (4.6 × 50 mm) column, with a gradient elution of ammonium acetate/acetic acid-acetonitrile as the mobile phase. Detection was performed by mass spectrometry (MS) in single ion recording (SIR) mode. Intra-day and inter-day precision data for both analytes were 3.8–7.2% and 3.6–7.4%, respectively. Calibration curves were both linear between 2 and 400 ng/mL with a correlation coefficient higher than 0.998. The LC-MS method can be used to quantify the drugs in human plasma in routine analysis. The method proved to be useful in determining real plasma levels in patients involved in cancer therapy. Drug concentrations were determined in a 10 min run-time, including re-equilibration of the column.

## 1. Introduction

Protein kinase inhibitors represent a large class of drugs used for therapy in cancer. Their use in anti-cancer treatment has grown significantly in the last years [1]. Among these inhibitors, drugs that interact with the cyclin-dependent kinases (CDKs) that control the cell cycle proved to be very promising antitumoral molecules. Although first CDK generation of drugs was toxic for non-tumor cells and failed in therapy, the second generation (the CDK 4/6 selective molecules) was found to be promising in clinical tests [2,3,4]. Since 2015, first palbociclib followed by abemaciclib and ribociclib were introduced in the US market. The specific inhibition of CDK 4/6-cyclin D complex presents the advantage of arresting cell-cycle between “G” and “S”-phase [5].

Bioanalysis of these compounds is gaining more and more interest. Later, with the development of mass spectrometry, smaller laboratories have begun to use highly sensitive detectors (including triple quadrupole mass analyzers), which are more accurate in determining chromatographic peak assignment and more sensitive in analyzing low plasma concentrations (usually ng/mL levels). Therefore, information on therapeutic drug monitoring (TDM) has become more accessible [6,7].

Determination of CDK 4/6 inhibitors concentration levels require an appropriate pre-treatment of plasma. Interfering plasma compounds such as phospholipids, proteins, salts, and exogenous substances may alter the electron ionization in MS and/or the chromatographic separation of target molecules [8]. Therefore, an appropriate sample pre-treatment method which provides at the same time a good “sample clean-up” and a high extraction recovery in human plasma is required. So far in the literature, authors usually used a simple protein precipitation as pre-treatment of samples in CDK inhibitors analysis [5,9,10,11]. To our knowledge, only Chavan et al. [12] used combined protein precipitation and SPE (with C18 cartridges) to extract these compounds from biological fluids. The main problem in protein precipitation is the incompatibility of the resulted supernatant with ESI source in MS detection, because of large amounts of co-extracted endogenous compounds. Liquid–liquid extraction usually generates clean samples, but requires a tedious working procedure, usually with multiple extraction steps and uses, also, large amounts of solvents. In SPE, low solvent and sample volumes are used and, in most cases, an even better sample cleanup compared to protein precipitation is performed.

All mentioned authors use UHPLC as the separation method. Detection is performed in all cases by tandem-mass spectrometry (MS/MS) or even QTOF-MS/MS [12]. MS/MS (especially QTOF-MS/MS) as detection methods are very sensitive and accurate, but also involve instruments that, in most cases, are not available to small laboratories or clinics.

The aim of this work was to investigate the extraction process of two CDK 4/6 inhibitors, palbociclib and abemaciclib in order to obtain a clean extract of the drugs. We developed and validated an accurate LC-MS method (which involves a more affordable instrumentation than MS/MS) for determining palbociclib and abemaciclib in human plasma. Solid phase extraction (SPE) using Oasis PRiME HLB^®^ cartridges was used for plasma sample preparation. The purpose was to optimize the cleaning procedure, therefore, to avoid the problems that may occur in ESI-MS electrospray interface. MS parameters were also assessed obtaining the best signal-to-noise ratio for both compounds. The method was fully validated and applied to real plasma samples.

## 2. Results

### 2.1. Validation Data

Figure 1 presents typical chromatograms of two standard samples in SIR mode, for the two analytes, with retention times of 4.32 and 4.89 min for palbociclib and abemaciclib, respectively. Both peaks were identified by spiking the final extracts with standard working solutions. Regarding selectivity, at specified *m*/*z* values (448 and 507), no other peaks are observed on the chromatogram. Chromatograms of palbociclib and abemaciclib in real plasma samples are presented in Figure 2.

Regarding the study of matrix effect, chromatograms obtained following the analysis of blank human plasma samples did not reveal interfering peaks at the above retention times and *m*/*z* values.

Calibration curves for both compounds are presented in Figure 3. Y axis presents peak area of the analytes; X axis presents plasma levels for palbociclib and abemaciclib. Correlation coefficients demonstrate the curves are linear in the selected concentration range.

The extraction recovery was, for both compounds and for all concentrations, more than 85%. LLOQ was 2 ng/mL. Table 1 reveals the stability tests results for palbociclib and abemaciclib. Long-term stability, freeze–thaw, and room temperature stability were higher than 95%.

Intra-day and inter-day precision were between 3.6 and 7.4%. All data for accuracy and precision are presented in Table 2.

### 2.2. Plasma Levels of Palbociclib and Abemaciclib

Plasma levels of palbociclib and abemaciclib in patients TDM were obtained using the linear regression equations. Palbociclib levels for the five patients were 65.6, 90.2, 97.5, 130.2, and 201.8 ng/mL. Calculated values for abemaciclib plasma levels in four patients were 8.1, 27.5, 29.9, and 56.2 ng/mL.

## 3. Discussion

A very important step in the development of an analytical method is the assessment of sample preparation process. Protein precipitation generates a supernatant usually incompatible with MS detection. LLE is tedious and pollutes the environment with large amounts of solvents. With this purpose, after performing proper investigations, we obtained the best results in sample preparation by SPE using Oasis PRiME HLB cartridges (Waters, Bucharest, Romania). The cartridges contain a water-wettable polymer capable of hydrophilic interactions with polar compounds. These cartridges use a very simple working protocol. The lack of conditioning and equilibration steps reduces solvent consumption and time analysis also. According to the manufacturer, a “three step clean-up” protocol was employed. In the first step, the sample (100 µL) is applied with a positive pressure to the cartridge and absorbed on sorbent surface, followed by washing step with 500 µL solution 5% methanol and elution step with 500 µL methanol. The use of acetonitrile in elution step is not recommended because of low solubility of CDK inhibitors in this solvent. Clean sample extracts were obtained by elimination of most interfering plasma matrix compounds, especially proteins and phospholipids.

After SPE, a concentration of the eluted solution was necessary to increase sensitivity of the method. The solvent (methanol) was evaporated in nitrogen stream and the residue was redissolved in 50 µL mobile phase, resulting in 10-fold increase in analytes concentration.

HPLC parameters were also investigated. A reverse phase CORTECS C18 chromatographic column (Waters, Bucharest, Romania) was used for separation. Acetonitrile/formic acid 0.1% aqueous solution as mobile phase produced important peak tailing, especially for abemaciclib. Ammonium acetate/acetic acid buffer 10 mM provided improved peak shape and acceptable retention times within 10 min run-time analysis. Mobile phase flow rate was adjusted to 0.8 mL/min and best resolution was obtained at 40 °C column temperature.

MS conditions were optimized. Positive ionization mode was employed. Capillary voltage and cone voltage were assessed to obtain the best signal for both analytes. Optimal values were 0.8 kV and 25 V for capillary and cone voltage, respectively.

Stability tests data (>95%) revealed no significant degradation of the analytes occurred during stock or sample preparation. Calibration curves were linear (correlation coefficient > 0.998). Precision (expressed as RSD) was lower than 10% (7.4% for ruxolitinib at LLOQ). Accuracy data were 95–110% to the nominal (spiked) concentrations.

The method was applied for the analysis of real plasma drugs levels. All blood samples were collected on day ten from the beginning of the treatment for drug levels to reach to the “steady-state” concentrations. Collection of samples was performed at two hours and one hour post dose respectively, representing the mean time (T_max_) to reach the maximum concentration in plasma (C_max_) according to the literature [4].

All detected plasma concentrations were in the 2–400 ng/mL range for both analytes. In real plasma samples, as presented so far in the literature, palbociclib and abemaciclib plasma levels also presented a large interindividual variability. This is a result of complex factors, starting with absorption and metabolism rates which depend on type of cytochrome P450 izo-enzymes, administration concerning the food and other drugs (enzymatic inhibition or induction) or patient’s adherence to the treatment.

Maximum plasma concentration for palbociclib and abemaciclib was observed 2 h after administration of the dose and it varied between 65.6–201.8 ng/mL and 8.1–56.2 ng/mL for each drug, respectively. These data demonstrate the large interindividual variability regarding abemaciclib and palbociclib pharmacokinetics, making TDM necessary in individual dose adjustment.

## 4. Materials and Methods

### 4.1. Materials

Palbociclib and abemaciclib were purchased from Sigma-Aldrich (Taufkirchen, Germany) and Alsachim (Illkirch Graffenstaden, France), respectively. Ammonium acetate, acetic acid, and MS grade ultrapure LC solvents (water, acetonitrile, methanol) came from Merck (Bucharest, Romania). Oasis PRiME HLB cartridges were purchased from Waters (Romania). Blank human plasma was achieved from our local blood bank (Blood Transfusion Center, Craiova, Romania). Chemical structures of palbociclib and abemaciclib are presented in Figure 4.

### 4.2. Methods

#### 4.2.1. LC-MS Chromatographic System and Analytical Parameters

The separation was carried out on a Waters (Milford, MA, USA) Arc System coupled with a Waters QDa mass detector. The column used was a Waters CORTECS C18 (4.6 × 50 mm, 2.7 μm particle size). Elution was carried out using solvent A (ammonium acetate/acetic acid 10 mM aqueous buffer solution) and solvent B (acetonitrile). The following gradient was used for analytical separation of the compounds: 0 min, 90% (A) and 10% (B); 0–8 min, gradually increasing eluent B to 60%; 8–9 min, maintaining B to 60%; 9–10 min, gradually decreasing eluent B to 10%. The flow rate of the mobile phase was set to 0.8 mL/min. The column was equilibrated at 40 °C temperature. The injection volume was 5 μL. All samples were kept at 20 °C in the autosampler during the entire analysis.

Eluted compounds were analyzed using a QDa mass detector (Waters, Bucharest, Romania) equipped with an electrospray ionization (ESI) source. Capillary voltage was maintained at 0.8 kV, cone voltage was kept at 25 V and the mass spectra were recorded in positive ion mode in the range *m*/*z* 100–500 for palbociclib and *m*/*z* 100–600 for abemaciclib. Quantification was performed in SIR mode for palbociclib and abemaciclib at *m*/*z* 448 and 507, respectively. The equipment was controlled using the EmPower 3 software package.

#### 4.2.2. Preparation of Stock and Working Standard Solutions

Stock solutions of palbociclib and abemaciclib at a concentration of 1 mg/mL were prepared in methanol. Palbociclib and abemaciclib mixed working solutions were prepared by diluting these solutions with methanol, to obtain final concentrations of 0.02, 0.2, 1, 2, and 4 µg/mL. All standard solutions were kept at −20 °C and renewed every 30 days.

#### 4.2.3. SPE Procedure for Standards and Samples

An aliquot of 10 µL mixed working solution was evaporated to dryness with nitrogen in a conical glass vial and 100 µL blank human plasma was added. Then, the mixture was loaded on Oasis PRiME HLB cartridges by using a positive pressure. The washing step consisted of 500 µL water with 5% methanol and analytes elution step was performed with 500 µL methanol. The elution solvent was evaporated to dryness and the residue was redissolved in 50 µL mixture consisting of acetonitrile: ammonium acetate/acetic acid 10 mM, 10:90 (*v*/*v*) representing the initial percentage ratio of the mobile phase. The solution was put in the autosampler and a volume of 5 µL was injected into the LC column.

#### 4.2.4. Validation Data

Validation was carried out according to International Council for Harmonisation (ICH) guidelines. Calibration curves concentrations obtained as described above for palbociclib and abemaciclib were 2, 20, 100, 200, and 400 ng/mL. Both intra-day and inter-day accuracy (percentage ratio between mean found concentration and spiked concentration) and precision (relative standard deviation—RSD%) of the analytes for all calibration levels were determined (n = 5).

Extraction recovery was also investigated for all concentrations. It represents the percentage ratio between the peak area of the extracted compound and the peak area resulted after direct injection of the same amount of drug dissolved in the initial mobile phase.

Lower limit of quantitation (LLOQ) is defined as the lowest concentration of the analytes on the calibration curve. LLOQ is accepted as limit of quantitation (LOQ) if the lowest standard has a signal to noise (S/N) ratio at least 10:1 and precision is no more than 10%. At the LLOQ, inter-day RSD was 7.3 and 7.4% for palbociclib and abemaciclib, respectively and the S/N ratio was 10:1; S/N ratio was the final criteria in establishing LLOQ.

Stability of the analytes was also investigated. Long-term stability (30 days at −20 °C) and freeze–thaw stability after three freeze-thaw cycles were established (n = 5). Stability at room temperature (20 °C) was determined for 12 h (n = 5) to detect the possible degradation of the analytes during sample preparation or in the autosampler, prior to injection.

Carry-over effect was tested by injecting a blank sample after the injection of standards containing both analytes at 400 ng/mL (maximum concentration of the calibration curve). The peak areas for the analytes in the blank sample should be less than 10% from the LLOQ peak area. The needle injection was automatically washed (ten times) with mobile phase (initial percentage ratio) between consecutive injections.

#### 4.2.5. Human Plasma Samples Analysis

To evaluate application of the method to real plasma samples, palbociclib plasmatic levels were investigated in five chronic patients treated with 75 mg palbociclib (Ibrance^®^) daily and in five chronic patients treated with 100 mg (50 mg/12 h) abemaciclib (Verzenios^®^) daily. Samples were collected two hours after administration of the daily dose (the morning dose in case of abemaciclib). Two milliliter of blood was collected into a vacutainer containing potassium EDTA, followed by centrifugation at 10,000 rpm for 5 min. Obtained plasma samples were kept at −20 °C before analysis.

Prior to analysis, all samples were thawed at room temperature and extracted as described in SPE procedure. All patients have given the informed consent regarding acquisition of plasma samples.

## 5. Conclusions

A simple LC-MS method was developed and validated for precise and accurate quantification of two CDK inhibitors in human plasma. The method required only 100 µL human plasma. The proper sample preparation (using a SPE technique without cartridge conditioning) produced clean extracts providing high sensitivity (LLOQ 2 ng/mL) for both compounds. The method was applied in determining palbociclib and abemaciclib plasma concentrations to nine patients undergoing treatment with these drugs. The novelty of this method consists in the simple three steps SPE sample preparation procedure and in the use of a simple MS detection technique, affordable to any laboratory. Validation data and results obtained on real plasma samples qualify this method to be used for determination of the analytes in human plasma.

## Figures and Tables

**Figure 1 molecules-27-08604-f001:**
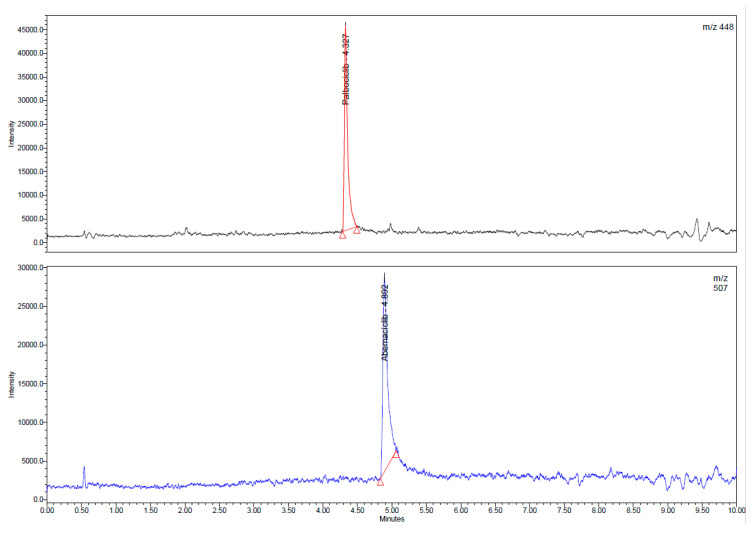
Chromatograms (SIR mode) for standard samples of palbociclib (200 ng/mL) and abemaciclib (100 ng/mL), with retention times of 4.32 and 4.89 min, respectively.

**Figure 2 molecules-27-08604-f002:**
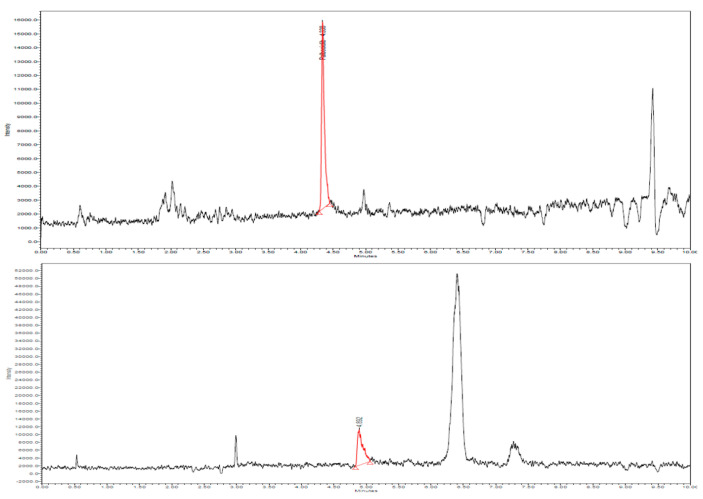
Chromatograms (SIR mode) for real plasma samples of palbociclib and abemaciclib, with calculated concentrations on the calibration curves 65.6 and 29.9 ng/mL, respectively.

**Figure 3 molecules-27-08604-f003:**
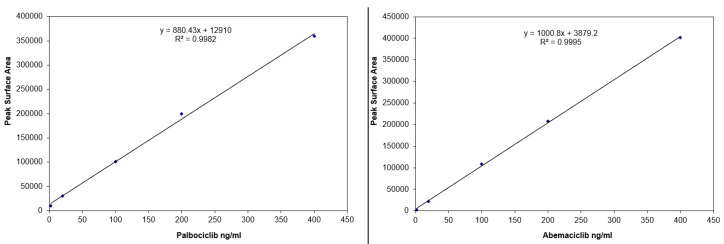
Calibration curves, linear regression equations, and correlation coefficients for palbociclib and abemaciclib.

**Figure 4 molecules-27-08604-f004:**
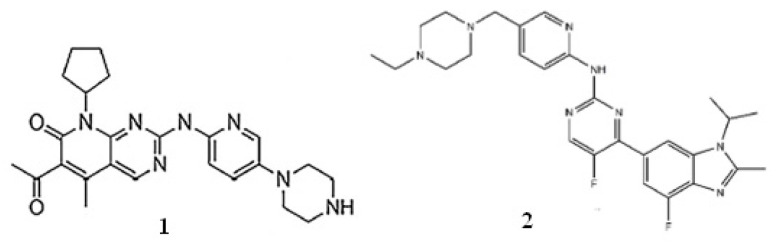
Chemical structures of palbociclib (**1**) and abemaciclib (**2**).

**Table 1 molecules-27-08604-t001:** Stability tests for palbociclib/abemaciclib (mean ± SD and percentage, n = 5).

Target Concentration (ng/mL)	Long Term Stability	Freeze-Thaw Stability	Room Temperature Stability
2	1.92 ± 0.03 (96.1%)/ 1.93 ± 0.03 (96.4%)	1.90 ± 0.04 (95.2%)/ 1.89 ± 0.04 (94.8%)	1.91 ± 0.04 (95.9%)/ 1.90 ± 0.04 (95.1%)
20	19.34 ± 0.32 (96.7%)/ 19.22 ± 0.37 (96.1%)	19.58 ± 0.20 (97.9%)/ 19.42 ± 0.37 (97.1%)	19.14 ± 0.41 (95.7%)/ 19.10 ± 0.42 (95.5%)
100	97.9 ± 1.02 (97.9%)/ 97.2 ± 1.36 (97.2%)	98.0 ± 0.98 (98.0%)/ 97.7 ± 1.12 (97.7%)	97.5 ± 1.21 (97.5%)/ 96.4 ± 1.73 (96.4%)
200	194.4 ± 3.69 (97.2%)/ 194.8 ± 2.53 (97.4%)	196.4 ± 1.76 (98.2%)/ 196.2 ± 1.86 (98.1%)	196.6 ± 1.67 (98.3%)/ 196.0 ± 1.96 (98.0%)
400	392.4 ± 3.72 (98.1%)/ 392.8 ± 3.53 (98.2%)	393.2 ± 3.34 (98.3%)/ 394.0 ± 2.95 (98.5%)	394.0 ± 2.95 (98.5%)/ 394.8 ± 2.56 (98.7%)

**Table 2 molecules-27-08604-t002:** Precision and accuracy data (n = 5).

Analyte	Target Concentration (ng/mL)	Mean Found Level (ng/mL)		RSD %		Accuracy %	
		Intra- Day	Inter- Day	Intra- Day	Inter- Day	Intra- Day	Inter- Day
Palbociclib	2	2.1 ± 0.1	2.2 ± 0.1	7.2	7.3	105.0	110.0
	20	20.4 ± 1.4	20.6 ± 1.2	6.8	6.5	102.0	103.0
	100	98.9 ± 5.6	98.0 ± 5.8	5.7	5.9	97.8	98.0
	200	202.5 ± 8.3	202.9 ± 9.5	4.1	4.7	101.2	101.4
	400	397.4 ± 15.1	395.2 ± 15.4	3.8	3.9	99.35	98.8
Abemaciclib	2	1.9 ± 0.1	2.1 ± 0.1	7.2	7.4	95.0	105.0
	20	19.6 ± 1.2	19.4 ± 1.2	6.8	6.5	98.0	97.0
	100	102.6 ± 6.0	103.0 ± 6.2	5.9	6.0	102.6	103.0
	200	197.7 ± 8.3	198.1 ± 10.3	4.2	5.2	98.8	99.0
	400	402.7 ± 15.3	403.6 ± 14.5	3.8	3.6	100.6	100.9

## Data Availability

Not applicable.

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
