# Peer review of "Development of a SPE-LC-MS Method for the Quantitation of Palbociclib and Abemaciclib in Human Plasma"

_molecules, 2022, doi:10.3390/molecules27238604_

Round 1

Reviewer 1 Report

The submitted manuscript tends to describe the development and validation of a method for the determination of palbociclib and abemaciclib in human plasma, using SPE-LC-MS.

The work is well-structured, clear and well-written. The novel aspect of this work lies on the use of SPE for sample preparation, while the existing methods use simple pretreatment with protein precipitation.

I suggest the authors to address the following issues;

1.       My major objection is that the study of matrix effect in the validation of the method is missing. Is it possible to analyze plasma samples using MS, without previous study of the matrix interferences?

2.       Line 206: The Lower Limit of Quantification (LLOQ) is defined as the lowest standard on the calibration curve, while the LOQ is 10 times Signal to Noise ratio (S/N). Please correct.

3.       Precision should be expressed as RSD%. Please correct it in Table 2, Line 136 and Line 201.

4.       The working solutions are prepared at levels 0.02, 0.2, 1, 2 and 4 µg/mL, which are equivalent to 20 – 4000 ng/mL. However, the range of 2 – 400 ng/mL is referred in Figure 2, Table 2, Line 143, Line 199, Line 214, and Line 233. Please correct.

5.       Ten column volumes are required to equilibrate the column, which corresponds to 5.8 mL of mobile phase and 7.3 minutes of equilibration (according to the column dimensions and the flow rate of the proposed method). Therefore, the 1 minute used (Line 173) is too short.

6.       In Figure 1, the concentration levels of the analyte are missing.

7.       In Figure 2, the measurement units are missing from the axes.

8.       The authors should add two chromatograms from the determination of the analytes in real samples.

Reviewer 2 Report

The main purpose of the work is to develop a new method for simultaneous determination of palbociclib and abemaciclib in human plasma. Then applied the method in TDM. The following should be revised

-          147 page 7…. enzymes, co-administration of food…… please change the expression of administration concerning the food.

-          Nothing mentioned about LLOD. Define LLOQ and how you obtained it

 -          Results of stability in table 1 should be mention in the form Mean ± SD in addition To the percentage

Round 2

Reviewer 1 Report

The authors prepared a fair revision of the original submission.
In my opinion, the revised version can be accepted in its current form.

Author Response

Dear Reviewer,

Thank you for performing revision of the manuscript!

Kind regards!

Manda Costel-Valentin,

Faculty of Pharmacy, University of Medicine and Pharmacy, Craiova

Reviewer 2 Report

In 2.1. Paragraph of validation data, the authors mentioned that “Regarding the study of matrix effect, chromatograms obtained following the analysis of blank human plasma samples did not reveal interfering peaks at the above retention times and m/z values”.  This is the wrong definition for matrix effects. Kindly see the paragraph 3.2.3. Matrix effect,  page 9 (ICH guideline M10 on bioanalytical method validation Step 5 (europa.eu)). Please add the results of the matrix effect of this method.

In 4.2.4. Validation data, Author mentioned that “validation was carried out according to International Council for Harmonization (ICH) guidelines. Validation is an essential part in the analytical method development. However, in the list of references, no mention of ICH guidelines.
